# Flexible and modular latent transition analysis—A tutorial using R

**Lisbeth Lund, Christian Ritz**◉*

National Institute of Public Health, University of Southern Denmark, Copenhagen K, Denmark

* ritz@sdu.dk

## Abstract

Latent transition analysis (LTA) is a useful statistical modelling approach for describe transitions between latent classes over time. LTA may be characterized in terms of prevalence at each time point and through transition probabilities over time. Investigating predictors of these transitions is often of key interest. Currently, LTA can mostly be carried out using commercial and specialized software and only to some limited extent by means of open source statistical software. This tutorial demonstrates a flexible and modular approach for LTA, providing a powerful alternative using **R** through a combination latent class analysis and multiple logistic regression models. This approach has several advantages from a modelling perspective, as demonstrated through revisiting a previously conducted LTA, published in PLoS ONE recently. In short, results were very similar to the original analysis using commercial software although some additional novel results were also obtained. The proposed alternative approach offers more options in terms of choice of effect measures, model assumptions such as hierarchical structures and covariate adjustment, and differential handling of missing data. R code snippets are provided in the tutorial. A detailed accompanying script is also provided for full reproducibility.

## Introduction

A range of statistical methods may be used for estimation of data-driven, exploratory identification of latent classes in longitudinal data. Statistical methods may crudely be divided into 1) ones that focus on characterization of determinants of longitudinal patterns and 2) ones that focus on characterization of determinants of transitions [1]. The fundamental distinction between the two approaches is whether individuals can be assumed 1) to remain in an identified latent class during the study period, that is a longitudinal pattern, or 2) to transition between latent classes over time during the study period.

The former is often referred to as estimating growth curves or latent class trajectories [2]. Typically these methods involve two steps: initial estimation of a number of trajectories, often by means of latent class analysis (LCA), followed by statistical modelling where membership of trajectories as an outcome is described by relevant determinants. One example from type II diabetes research is the use of regression modelling to characterize obesity patterns by means of available baseline information; another example is characterization of child growth using

**Data Availability Statement:** All relevant data are provided within the manuscript and the supplementary material on zenodo.org (https://doi.org/10.5281/zenodo.10794077).

**Competing interests:** The authors have declared that no competing interests exist.

baseline information [3, 4]. Combined estimation of the latent class and mixed model structures in one step is also possible and it has been used for modelling cerebral and functional aging [5]. In short, these methods are powerful for characterizing features that are associated with certain trajectories.

The latter approach, which is the topic of this tutorial, does not focus on characterizing static pattern membership but on dynamic transition between patterns as time evolves. Such a characterization makes sense in case the longitudinal data correspond to having observed participants at several distinct and well-defined time points that are well-separated in time, as is common in randomized trials and, more generally, in some types of cohort studies. The corresponding statistical method is called latent transition analysis (LTA). In contrast to above LCA-type methods where pattern membership of each participant is determined once and assumed constant over time, LTA allows for pattern membership to change over time, capturing the dynamics in participants' behaviour over time. LTA has been applied to explore and explain transitions in substance, tobacco, and nicotine use, changes between dietary patterns and between psychological needs profiles [6–9]; several of these examples are secondary analyses of data from randomized controlled trials where the interaction between transitions and interventions was of interest.

Currently, mostly highly specialized commercial statistical software is used to carry out LTA, including *Latent GOLD*® with the *Adv/Syntax* add-on (https://www.statisticalinnovations.com) and *Mplus* (https://www.statmodel.com) [10]. A freely available add-on procedure is available for SAS [11]. These software solution rely on fitting multinomial models. There is also similar functionality in **R** for carrying out LTA [12]. Specifically, the package "LMest" can be used for estimating transition probabilities and also for investigating how transitions are affected by covariates, assuming a multinomial regression model [13]. There also exists an **R** package "MplusAutomation" providing convenience functions for using with the above-mentioned software *Mplus*.

The aim of this tutorial is to demonstrate that LTA can be carried out in a very modular way through the open source statistical environment **R**. This approach involves a combination of modelling steps, including latent class analysis and logistic regression. Moreover, this approach also provides a much greater flexibility in terms of model specification, modelling assumptions, and handling of missing values, as compared to existing software solutions that rely on multinomial models. **R** code is shown for the re-analysis of published open-access data, previously analyzed using commercial software.

## Materials and methods

### Data

To study well-being in early adolescence and, specifically, to explore how well-being changes over time, a longitudinal study with two waves in autumn 2019 and 2020 was carried out in Switzerland [14]. At each wave the participants were asked to fill in an online questionnaire. In order to construct latent classes, six indicators were used as detailed below. Specifically, through the online questionnaire, the participating high school students were asked about three hedonic indicators (life satisfaction (range 1 to 7), self-esteem (range 1 to 4), and well-being (range 1 to 5)) and three eudemonic indicators (self-efficacy, self-determination, and satisfaction with grades at school (range 1 to 4 for all three indicators); these indicators were based on validated scales. For each indicator, Likert scales were originally used, but only dichotomized versions (low vs. high) were available in the accompanying open-access dataset. In addition, a number of potential predictor variables were recorded: age, gender, socio-economic status (SES) based on information on parental education level (0 to 5, with 0 representing the highest education level

and 5 the lowest), migration background (yes or no), and school level based on academic performance and teacher recommendations (low or high) [14].

## Latent transition analysis

In principle, LTA may be carried out in a single estimation step assuming a simultaneous statistical model including LCA (for a fixed number of classes). Subsequently transition probabilities and odds ratios are estimated [13]. However, LTA is often carried out in two separate steps: LCA and multinomial regression. In this tutorial, an alternative approach is proposed where the multinomial regression model has been replaced by multiple logistic regression models, leading to three separate analytic steps, which can conveniently handled sequentially, one at a time.

**Step 1.** In the first step, LCA was applied to each wave separately to identify distinct latent classes that capture salient features in the indicator variables. In short, LCA corresponds to fitting a mixture model, assuming that the data follow a distribution consisting of a mixture of several normal distributions, interpreted as the underlying distributions that generate the latent classes. Estimation aims to find the model parameters that assign the highest likelihood to the data (the global maximum of the likelihood function is being searched for), and it is commonly carried out for a fixed number of latent classes, which needs to be specified in advance. However, multiple models for different choices of the number of latent classes may easily be fitted and subsequently compared. In our analysis, LCA models were fitted assuming 2 to 5 latent classes [15]. To avoid ending up with sub-optimal model fits due to local maxima found during the estimation, which is not uncommon to LCA, one approach is to estimate the LCA models multiple times with different starting values for the parameter estimates [15]. Trying out a range of starting values should be done routinely when fitting LCA models.

The optimal number of latent classes is often decided on using an information criterion, such as Akaike information criteria (AIC) and Bayesian information criteria (BIC), or adjusted versions of AIC and BIC [16]. Following the approach of the original study, AIC and the sample-size adjusted BIC were reported [17]. A difference in an information criterion less than 10 means that the difference between the two model fits is negligible [18]. However, other considerations, such as the interpretability of results, may also play an important role in deciding on the optimal number of latent classes [2, 9]. LCA produced prevalences of identified latent classes and item-response probabilities (conditional on a latent class). Item-response probabilities denote the likelihood of responding yes/no to a specific indicator variable, given the latent class status and they were used to distinguish between latent classes and, consequently, to assign a meaningful label to each latent class. In addition, item-response probabilities at wave 1 and 2 were compared visually to assess measurement invariance across time, i.e., similar probabilities would support the measurement invariance assumption that the same latent class structure was observed both at wave 1 and 2 [6]. As measurement invariance is a modelling assumption, a visual assessment seems sensible and in line with common practice for other statistical models such a linear regression; statistical tests for model assumptions should be avoided and an element of subjective judgment should be accepted [19, 20]. In case the assumption of measurement invariance is not tenable then the proposed approach is still applicable. The interpretation of results may, however, become more challenging (but in some cases most likely also more realistic) as it would mean that individuals transition between different classes over time. For instance, it might be difficult to entertain measurement invariance for large gaps between waves.

In Steps 2 and 3, as detailed below, multiple logistic regression models were fitted as a flexible alternative to multinomial regression models. It has been shown that there is almost no loss in efficiency when using logistic regression instead of multinomial regression [21].

**Step 2.** Logistic regression models were used to estimate transition probabilities. Specifically, a binary outcome variable was defined for each class at wave 2, indicating whether or not the participant belongs to that particular class. Separate logistic regression models were fitted for each of these binary outcomes, with class membership at wave 1 as the only independent (categorical) variable. The number of logistic regression models that were fitted corresponded to the number of classes at wave 2. As results from logistic regression, by tradition, are reported on a logarithmic scale, a subsequent back-transformation step was needed. Specifically, back-transformation from the log-odds scale was used to obtained estimated transition probabilities for changing from one latent class at wave 1 to any of the three latent classes at wave 2; corresponding 95% confidence intervals were also obtained through back-transformation.

**Step 3.** Logistic regression models were fitted to estimate ORs that quantify associations between transitions from wave 1 to wave 2 and relevant predictors. Specifically, for each transition of interest, a binary outcome variable was defined indicating whether or not the participant underwent the transition or remained in the same class at wave 2 as at wave 1. Separate logistic regression models were fitted for each of these derived binary outcomes and for each of the following five predictors: age, gender, migration, school, and socio-economic status. Each of these logistic regression models included an interaction term between the class membership at wave 1 and the predictor in question. The number of logistic regression models fitted for each predictor corresponded to the number of transitions of interest. In this tutorial, three transitions were considered: from low to middle wellbeing, low to high wellbeing, and middle to high wellbeing [14]. However, it should be noted that there would be three more transitions to consider (middle to low, high to low, and high to middle), but they seem less relevant in the present context as they corresponded to transitions towards reduced instead of improved well-being.

**Handling missing values.** Missing values for the latent class membership were handled using multiple imputation through chained equations (MICE) based on the six input variables at wave 1 and 2, predictors to be investigated (age, gender, socio-economic status, migration status, and school level), and the latent class membership variables at wave 1 and wave 2 (in total 19 variables). Based on this dataset, MICE was used to generate 10 complete, imputed datasets. Often a small number of imputed datasets, such as 5 or 10, are used as it suffices for capturing the uncertainty in the imputation step, but also avoids that the computational burden becomes too large [22]. Note that a pre-specified random seed was given to ensure reproducibility of results, as was also the case for LCA in the above Step 1. For each imputed dataset, separate model logistic regression models fits were obtained. Subsequently, individual parameter estimates were combined using Rubin's rule, which makes due allowance for the uncertainty introduced through use of multiple imputation [23]. The MICE approach is advantageous as it can handle arbitrary types of variables (e.g., continuous and categorical variables), ensuring that imputed values will be of the same type as the observed values [23]. In this tutorial MICE was applied after LCA but before any fitting logistic regression models as multiple imputation is useful for statistical inference, obtaining estimates and standard errors that reflect the uncertainty in the imputation step.

**Software used.** Statistical analyses were carried out in R using the extension packages "mice" (version 3.15) and "poLCA" (version 1.6) [12, 15, 23]. Supplementary material containing an annotated **R** script, including precise definitions of binary outcome variables used and detailed explanations of model specifications, is available online at zenodo.org (https://doi.org/10.5281/zenodo.10794077). Key steps using **R** functionality are also explained and shown in boxes in the Results section. A significance level of 0.05 was assumed.

**Approach in the original study.** LTA was carried out using the commercial software *Mplus*. Specifically, latent classes were identified and the corresponding transition probabilities were estimated. Missing values in wave 1 were imputed although the imputation method was not described. Subsequently, for each wave, a multivariate multinomial regression model was fitted with latent class membership as a multinomial outcome and gender, migration background (yes or no), school level, and socio-economic status as predictors. However, these analyses, which were carried out using SPSS (version 25), did not investigate the effects of predictors on transitions but only evaluated cross-sectional effects [14].

## Results

### Descriptive statistics for wave 1

A total of 377 students completed the questionnaires at both wave 1 and 2. There were 167 female and 190 male respondents, respectively, and 20 respondents missing gender information. The age range was 11–15 years, but almost all respondents were 12–14 years (99%); 13 respondents did not provide information on age. Socio-economic status was reported using five categories (lowest, lower, middle, high, and highest), with most respondents in the middle or lower categories (51%). Data on socio-economic status were missing for 58 respondents. Likewise, there were 13 missing values for the well-being indicator, which will be used to define latent classes, for wave 1.

**LTA step 1—estimating latent classes.** Box 1 shows the **R** code for carrying out LCA, exemplified for wave 1; it can briefly be explained as follows: the LCA requires that input variables are initially combined into a model formula, which will be the first argument for the function poLCA() from the extension package of the same name. The second argument of poLCA() needs to be the data set where the variables in the model formula can be found. Finally, the third argument specifies the number of latent classes to be assumed. The random number generator is initiated by means of the function set.seed() to ensure reproducibility.

Box 1. LCA at wave in R

```
library(poLCA) # activating the R package "poLCA"

# defining a suitable model formula

varList1 <- cbind(lifesat1, selfeff1, selfacc1, selfdet1, wellbe1, satis1) ~ 1

# fitting LCA models

set.seed(202408143)

lc1.2 <- poLCA(varList1, ltadata, nclass = 2, nrep = 10) # LCA assuming 2 classes

lc1.3 <- poLCA(varList1, ltadata, nclass = 3, nrep = 10)

lc1.4 <- poLCA(varList1, ltadata, nclass = 4, nrep = 10)

lc1.5 <- poLCA(varList1, ltadata, nclass = 5, nrep = 10)

# showing the results

lc1.2

lc1.3

lc1.4

lc1.5
```

**Table 1. Model fit statistics for the proposed approach and as reported in the original study by Kassis et al. (n = 377).**

| Proposed approach | | Wave 1* | | Wave 2 | |
|---|---|---|---|---|---|
| No latent classes | No. parameters | AIC | aBIC | AIC | aBIC |
| 2 | 13 | 2667 | 2677 | 2579 | 2589 |
| 3 | 20 | 2651 | 2665 | 2555 | 2570 |
| 4 | 27 | 2654 | 2673 | 2543 | 2563 |
| 5 | 34 | 2656 | 2681 | 2548 | 2574 |
| Original study | | Wave 1 | | Wave 2 | |
| No. latent classes | No. parameters | AIC | aBIC | AIC | aBIC |
| 2 | 13 | 2729 | 2738 | 2578 | 2588 |
| 3 | 20 | 2707 | 2723 | 2555 | 2570 |
| 4 | 27 | 2710 | 2730 | 2542 | 2563 |
| 5 | 34 | 2712 | 2738 | 2547 | 2573 |

*: n = 364 as there were 13 missing values for the well-being indicator.

Abbreviations: AIC = Akaike's information criterion, aBIC = sample-size adjusted Bayesian information criterion.

Based on the fitted LCA models, relevant fit statistics were extracted as summarized in Table 1. The model fit statistics indicated that either 2 or 3 latent classes were optimal. This result was found both for the proposed alternative approach and for the original study despite the discrepancy in AIC and aBIC values for wave 1 where missing values were imputed in the original study. Specifically, for both waves, AIC values did not appreciably differ for 3, 4, and 5 classes (as differences were not larger than 10) but AIC was appreciably larger for 2 classes. A similar picture was observed for aBIC. Consequently, a parsimonious choice was to select the smallest number of latent classes among the options with similar AIC and aBIC values, resulting in a three latent class solution; this choice also ensured that sufficient participants were included in each class [16]. The above analysis, as described in Box 1, included the argument nrep = 10 in the function poLCA(), implying that LCA was repeated 10 times with different starting values to ensure that the final model fit corresponded the global maximum and not some local maximum [15].

The three latent classes were well-separated as item response probabilities group in a low, middle, and high trend (Fig 1). Moreover, the same latent classes were found for both waves, supporting the measurement invariance model assumption [10]. For wave 1, predicted class membership percentages were 21%, 48%, and 31% for the low, middle, and high well-being classes, respectively, compared to 24%, 46%, and 30% in the original study; the small discrepancy was due the missing values being imputed in the original study. For wave 2, predicted class membership percentages were 31%, 31%, and 38% for the low, middle, and high well-being classes, respectively, compared to 31%, 30.5%, and 38.5% in the original study.

**LTA step 2—estimating transition probabilities.** In **R** multiple imputation is carried out using the package "mice" as shown in Box 2. The first argument of the function named mice() was the dataset containing the variables that should be used for the imputation step. In this case all variables needed for the subsequent logistic regression analyses were provided. The second argument specified that 10 complete, imputed datasets should be generated. Note that the seed for the random number generator is again specified using the function set.seed(). Once these datasets were constructed, logistic regression models were fitted to each of these 10 datasets using the function with() with arguments "with" and "exp", specifying the list of imputed datasets and the logistic regression model to be fitted. The logistic regression model is fitted using the model fitting function glm() with an outcome that is defined as being in high

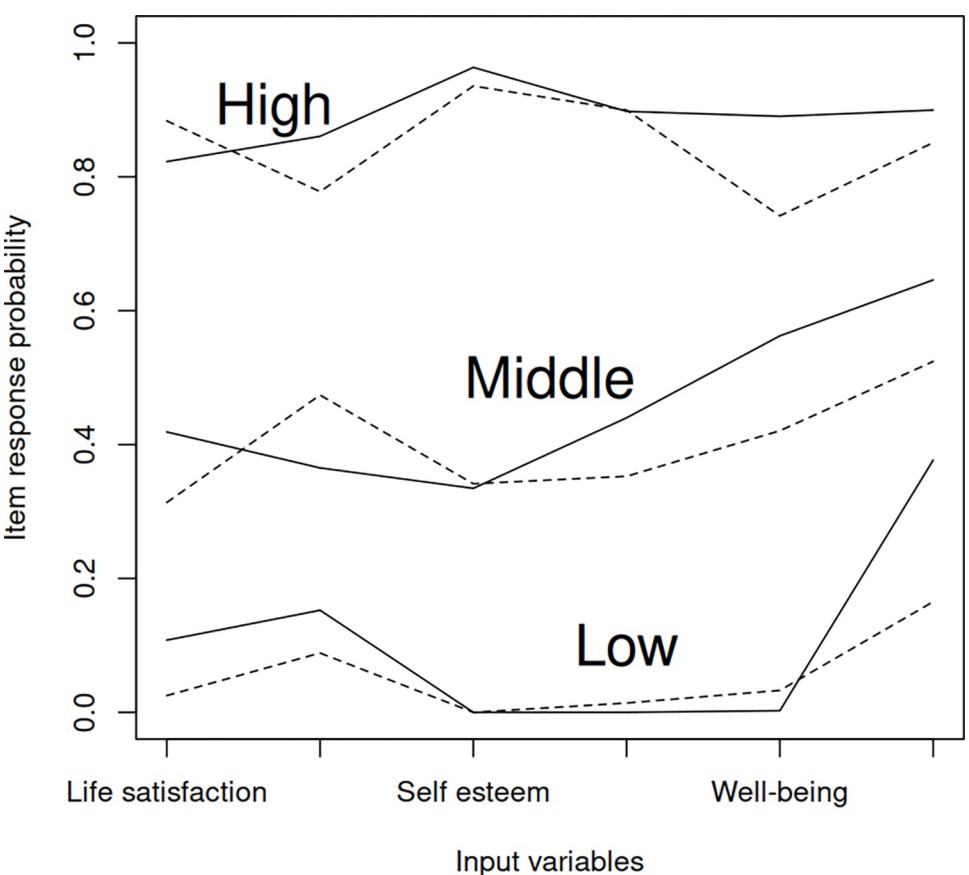

**Fig 1. Item-response probabilities for the low, middle, and high well-being latent classes.** Solid and dashed lines connect item-response probabilities between the 6 input variables (indicators) for wave 1 and wave 2, respectively.

well-being class at wave 2 (yes or no), a single predictor that is the variable denoting the class membership at wave 1, and, finally, the argument needed to inform glm() that a logistic regression model is specified: link = binomial, ie., logistic regression is a generalized linear model (glm) with a link function that accommodates a binomial distribution [24]. Subsequently, the use of summary(pool()) implied that results from the 10 separate model fits were combined into a single result, in this case three parameter estimates corresponding to the three latent classes at wave 1. These estimates were then back-transformed to probabilties.

---

Box 2. Transition probabilities in R

library(mice) # *activating the R package "mice"*

# *constructing a dataset only containing the variables to be imputed/used in the analyses*

# *lca1 and lca2 are variables on class membership at wave 1 and 2*

ltadata.sim <- cbind(ltadata[, c("School_Level", "Gender", "Age", "Migration", "Socioeconomicstatus", "lifesat1", "selfeff1", "selfacc1", "selfdet1", "wellbe1", "satis1",

"lifesat2", "selfeff2", "selfacc2", "selfdet2", "wellbe2", "satis2")], lca1, lca2)

# *generating 10 complete, imputed datasets*

---

```
set.seed(202401233)

ltadata.imputed <- mice(ltadata.sim, m = 10)

# fitting logistic regression to each imputed dataset

ltadata.mice1 <- with(data = ltadata.imputed,

        exp = glm (lca2 = = "high" ~ lca1–1, family = binomial))

# pooling results using Rubin's rule

logreg1 <- summary(pool(ltadata.mice1))

# transforming back to probability scale

coef1 <- logreg1[, 2]

llim1 <- coef1–1.96 * logreg1[, 3]

ulim1 <- coef1 + 1.96 * logreg1[, 3]

data.frame(class = logreg1[, 1],

        exp(cbind(coef1, llim1, ulim1)) / (1 + exp(cbind(coef1, llim1, ulim1))))
```

Both for the low and high well-being classes, the majority of students were most likely to remain in the same class in both waves (Table 2). For the high well-being class, there was a 67% probability of remaining in that class, whereas it was somewhat smaller, only 56% for the low well-being class. For the middle well-being class, transition to either low or high well-being classes at wave 2 was almost equally likely, with probabilities of 27% and 33%, respectively, corresponding to a 40% probability of remaining in middle well-being class.

**LTA step 3—investigating predictors of transition.** Box 3 shows the **R** code for fitting a logistic regression model where the effect of a specific predictor on a specific transition was investigated. In this case it was the effect of age on the transition from low to middle (or remain in low); for the other predictors the **R** code may be found in the online supplementary

**Table 2. Transition probabilities between wave 1 and 2 expressed as percentages (n = 377) for the proposedapproach and as reported in the original study by Kassis et al.**

| Proposed approach | Wave 2 class | | |
|---|---|---|---|
| Wave 1 class | Low well-being | Middle well-being | High well-being |
| Low well-being | 55.6 (44.9, 65.8) | 31.1 (22.0, 41.9) | 13.3 (7.6, 22.2) |
| Middle well-being | 27.3 (21.2, 34.4) | 40.1 (33.0, 47.6) | 32.6 (26.0, 40.0) |
| High well-being | 17.7 (11.7, 25.9) | 15.0 (9.6, 22.9) | 67.3 (58.1, 75.3) |
| Original study* | Wave 2 class | | |
| Wave 1 class | Low well-being | Middle well-being | High well-being |
| Low well-being | 54.4 | 32.2 | 13.3 |
| Middle well-being | 27.6 | 39.6 | 32.8 |
| High well-being | 17.7 | 15.0 | 67.3 |

Data are shown as estimated percentages and 95% confidence intervals (back-transformed from the log odds scale).

*: For the original study only estimated percentages without corresponding confidence intervals were reported.

material. As in Step 2 the logistic regression model was fitted to each of the imputed datasets. The logistic regression model was fitted to the subset corresponding to low or middle well-being at wave 2, and, consequently, the outcome, which was class membership at wave 2, became binary and suitable for logistic regression. Moreover, the model included an interaction between class membership at wave 1 and the predictor age. The model specification involved both the main effect of the class membership variable and the interaction term specified using ":" and not "*". This parameterization ensured that the output contained results that could be directly back-transformed into interpretable odds ratios reported in Table 3 below.

---

**Box 3. Predictors of transition in R**

*# fitting logistic regression to each imputed dataset*

*# with the transition to low or middle well-being (middle = 1; low = 0) at wave 2 as outcome*

*mice.step3 <- with(data = ltadata.imputed,*

 *exp = glm (lca2 = = "mid" ~ lca1 + lca1:Age-Age, family = binomial,*

 *subset = lca2%in% c("low", "mid")))*

*# pooling results using Rubin's rule*

*logreg.step3 <- summary(pool(mice.step3))*

*# back-transforming to OR for transitioning from low well-being at wave 1*

*# (row 6 corresponds to "low" at wave 1)*

*coef.step3 <- as.vector(unlist(logreg.step3[6,])) # row 6 selected*

*c(exp(c(coef.step3[2], # estimated OR*

 *coef.step3[2] - 1.96\*coef.step3[3], # lower limit of 95% CI*

 *coef.step3[2] + 1.96\*coef.step3[3])), # upper limit of 95% CI*

 *coef.step3[6]) # p-value*

---

There was a significant association between age and the low to middle transition, where the odds of transition were increased by 179% (95% CI: [27, 511] %; p = 0.01) per year, whereas odds for transitions from low to high and middle to high well-being class were reduced by 20% and 1% per year, respectively (Table 3).

No other associations between predictors and transitions were significant. Other notable findings included: Male students had higher odds for the transition from low to middle and low to high well-being classes compared to female students (64% and 29%, respectively) but 28% reduced odds for the transition from middle to high. Migration only affected transitions slightly (-14% to 2% change in odds). Students at higher secondary schools had 79% and 87% higher odds for transitioning from low to middle well-being and from low to high well-being, respectively, but 52% reduced odds for transitioning from middle to high well-being.

The high SES group had 20% and 10% higher odds for the transition from low to middle and from low to high well-being, respectively, compared to the middle SES group, but only 1%

**Table 3. Associations between selected transition and predictors (n = 377).**

| | Transition | | | | | |
| --- | --- | --- | --- | --- | --- | --- |
| | Low to Middle | | Low to High | | Middle to High | |
| Predictor | OR (95% CI) | p-value | OR (95% CI) | p-value | OR (95% CI) | p-value |
| Age | | | | | | |
| (per 1 year) | **2.79 (1.27, 6.11)** | **0.01** | 0.80 (0.25, 2.54) | 0.70 | 1.01 (0.62, 1.65) | 0.97 |
| Gender | | | | | | |
| (male vs. female) | 1.64 (0.31, 8.54) | 0.56 | 1.29 (0.24, 7.12) | 0.77 | 0.72 (0.20, 2.62) | 0.62 |
| Migration | | | | | | |
| (with vs. without) | 1.02 (0.19, 5.45) | 0.98 | 0.86 (0.16, 4.55) | 0.86 | 0.97 (0.27, 3.46) | 0.96 |
| School | | | | | | |
| (high vs. low) | 1.79 (0.31, 10.27) | 0.51 | 1.87 (0.32, 11.00) | 0.49 | 0.48 (0.13, 1.70) | 0.25 |
| SES | | | | | | |
| (high vs. middle) | 1.20 (0.05, 30.12) | 0.91 | 1.10 (0.08, 14.45) | 0.94 | 1.01 (0.12, 8.71) | 0.99 |
| (low vs. middle) | 0.22 (0.01, 4.03) | 0.31 | 0.45 (0.04, 4.56) | 0.50 | 4.65 (0.94, 23.01) | 0.06 |

Data shown are estimated OR's from logistic regression models that included an interaction between the latent class membership variable at wave 1 and the predictor in question. Significant findings are shown in bold.

Abbreviation: SES = Socio-Economic Status, OR = Odds Ratio.

higher for the transition from middle to high well-being. In contrast, the low SES group had 78% and 55% reduced odds for the transitions from low to middle and from low to high, respectively, but 465% increased odds for the transition from middle to high.

No similar results were reported in the original study where only cross-sectional associations of predictors of class membership at wave 1 and wave 2, respectively, were investigated [14]. It is also worth noting that the effect of age was not investigated at all in the original study. However, in the original study, a gender effect was found at wave 2 but not at wave 1.

## Discussion

The proposed alternative approach identified exactly the same three latent classes and found very similar estimated transition probabilities when compared to the results obtained using commercial software in the original study [14]. The findings in Step 3 could not be compared as they were not reported in the original study. However, this tutorial explicitly imputed missing values such that transition probabilities and OR's were estimated using data from all 377 students. Moreover, as a novel contribution and added value compared to the original study, associations between predictors and selected transitions were investigated and indeed resulted in establishing a novel and meaningful association between age and the transition from low to middle well-being; this finding may be a chance finding as any kind of multiplicity adjustment of the reported p-values would render it non-significant. Moreover, it should be noted that many LTA's are secondary analyses of data from studies that were neither designed nor powered to investigate transitions over time. Indeed, LTA may be viewed in the same way as statistical analyses investigating effect modifications (as Step 3 involves interaction terms); these are also often under-powered.

The proposed approach gains flexibility through a stepwise procedure that essentially replaces fitting a multinomial model by fitting a number of logistic regression models. Consequently. this approach has a number of advantages as compared to available software solutions: Arbitrary hierarchical study designs, such as cluster-randomized trials and repeated measurements studies, can be conveniently handled using logistic mixed-effects model with suitable

random effects, including random intercepts, which can be different for different models. Similarly, covariate adjustments can be easily included, and they may vary across analyses e.g., be different for different waves or even transitions. Missing values can be handled flexibly through imputation methods such as MICE [23]. In this tutorial, the same imputation model was used for all logistic regression models, but differential handling of missing values through different collections of imputed datasets would also be possible. Over-dispersion can be addressed and, more generally, robust sandwich-type standard errors may be applied in case of model misspecification [25]. In addition, relative risks and risk differences may be estimated as alternatives to odds ratios where appropriate [26]; this is not easily achieved using available LTA software, which relies on multinomial models with odds ratio as the effect measure. The proposed LTA approach does not require classes to remain consistent across waves. Admittedly, changing classes over time renders interpretation of transitions more difficult but possibly also more realistic, in particular if time gaps between waves are large. In contrast to the proposed approach, the **R** package "LMest" is of limited value for analyzing more complex study designs involving hierarchical structures and it lacks flexible handling of missing values and is limited to odds ratios as effect measures. Finally, as an alternative to **R**, the proposed LTA approach could also be carried out using the commercial, but general-purpose statistical software STATA (StataCorp, College Station, Texas 77845, USA) with the LCA Stata plug-in [27, 28]. The advantage of using **R** or Stata is that LTA can be carried out step-wise such that powerful functionality can be applied in each step, rendering the statistical analysis very flexible. Output from one step feeds directly into the next step.

The modular structure of the proposed approach also entails some limitations. In contrast to a single joint model for both the LCA step and subsequent estimation of transition probabilities and odds ratios, some uncertainty from the estimation of latent classes in Step 1 will not be propagated to Step 2 and Step 3. Specifically, class membership at wave 1 will be assumed to be known without error. One way to alleviate this problem is to consider modifications of logistic regression that address measurement error [29, 30] or using a weighting method [31], exploiting the modular structure of the approach even more. Some additional limitations should also be mentioned. There exist several alternatives to the **R** package "poLCA" for doing LCA, providing additional flexibility, such as the packages "BayesLCA" and "tidySEM" [32, 33]. On a related note, it should be pointed out that the use of LCA as outlined in Step 1 (using poLCA() in **R)** tacitly assumed independent data, which might be an assumption that is compromised by certain hierarchical structures. Extensions of LCA allowing for dependent data could be consider instead, including the **R** packages "glca" and "multilevLCA" [13, 34].

It has previously been shown that using multiple logistic regression models instead of a multinomial regression model leads to a very modest loss in efficiency, but it cannot be ruled out that a correctly specified multinomial regression model leads to a small gain in efficiency, especially in case of highly correlated binary outcomes [21]. Specifically, multinomial regression models could also be used in the proposed approach in Step 2 and 3 instead of logistic regression, but they introduce more modelling assumptions. However, the stronger assumptions also alleviate some of the problems encountered in case of sparse data where logistic regression models may face convergence problems. Another limitation is that LTA is only applicable to longitudinal data from randomized trials and repeated cross-sectional studies where time points often are well-defined and the same for all participants. However, LTA is not suitable for longitudinal data where measurements for each participant may be taken at different time points as is not unusual in many cohort studies. Finally, it is a limitation and indeed the price for a high degree of flexibility that the programming needed involves some lines of code as shown in Box 1, 2 and 3, although it should be mentioned that also *Mplus* and SAS, say, require programming in order to define suitable matrices prior to carrying out LTA [10].

## Author Contributions

**Conceptualization:** Christian Ritz.

**Methodology:** Lisbeth Lund, Christian Ritz.

**Software:** Christian Ritz.

**Validation:** Lisbeth Lund.

**Writing – original draft:** Christian Ritz.

**Writing – review & editing:** Lisbeth Lund, Christian Ritz.

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
