## [Decision Letter · Decision Letter 0]

6 May 2024

PONE-D-24-09372Flexible and modular latent transition analysis for subgroup identification and characterization using RPLOS ONE

Dear Dr. Ritz,

Thank you for submitting your manuscript to PLOS ONE. After careful consideration, we feel that it has merit but does not fully meet PLOS ONE’s publication criteria as it currently stands. Therefore, we invite you to submit a revised version of the manuscript that addresses the points raised during the review process.

We look forward to receiving your revised manuscript.

Kind regards,

Sandar Tin Tin

Academic Editor

PLOS ONE

Reviewers' comments:

Reviewer's Responses to Questions

**Comments to the Author**

1. Is the manuscript technically sound, and do the data support the conclusions?

Reviewer #1: Yes

Reviewer #2: Partly

Reviewer #3: Yes

2. Has the statistical analysis been performed appropriately and rigorously? 

Reviewer #1: Yes

Reviewer #2: No

Reviewer #3: Yes

3. Have the authors made all data underlying the findings in their manuscript fully available?

Reviewer #1: Yes

Reviewer #2: Yes

Reviewer #3: Yes

4. Is the manuscript presented in an intelligible fashion and written in standard English?

Reviewer #1: Yes

Reviewer #2: Yes

Reviewer #3: Yes

5. Review Comments to the Author

Reviewer #1: The authors describe an implementation of latent transition analysis (LTA) with the statistical software R, based on an example study about wellbeing of high school students previously published in 2022.

Instead of parameter estimation in a single step, the analysis is split into at least two steps, first estimating latent classes, and then estimating transition probabilities and the effect of predictor variables on these transitions in a second or third step.

Given that the latent class membership has been estimated in Step 1, there is uncertainty about whether a participant belongs to a particular class. The logistic regression models assume instead a fixed class membership for each participant. Is it possible that this assumption affects the results? Can the uncertainty about class membership be carried forward into Step 2, for example using weights based on the item-response probabilities?

Step 2 and Step 3 have been simplified by using separate binary logistic regression models instead of a multinomial regression model. If the correlation between classes is not important, this is a reasonable approximation, but the authors note in the discussion that multinomial regression models are a feasible alternative. I agree that this simplification also allows for modelling more complex sampling structures, although it might be more of a challenge to include them in the first LCA step.

In the R script, it looks as if separate models were fitted for each predictor variable. Can we instead assume a single model conditioning on the full set of predictors?

As this is a replicated analysis of the Kassis et al. 2022 study, would it be possible to show the differences to the previous analysis in the Results Section?

Minor issues:

Page 8: There is some functionality for carrying out LTA is available in R [12].

Instead: There is some functionality available in R [12] for carrying out LTA.

Page 8: Missing empty space: Likert scales were originally used, butonly dichotomized versions

Checking capital letters for names, for example, Wave 1 and 2

Reviewer #2: The title is slightly misleading. Subgroup is usually used to refer to groups of subjects, that is the group of subjects that the subject is in for all time, which in this case there is only one. Then the subject may change class at each time point..

The introduction implied that the paper was going to provide a tutorial on using the LMest package to fit LTA models. Although it may not be suitable for PLOS, this would be useful, as it is a model that is not covered in the authors description of the LMEst package. (Bartolucci et al, 2017). Note that it is a simplified model to that shown in their 6.2, with only one class for individuals, so may be fitted with lmest.

However they show how to fit these models in a 3 step process. I think most statisticians would find this unappealing. When determining the latent classes, this requires them to be determined at each wave separately, rather than having a common set of classes, which are applied at all waves. The major difficulty is that rather than considering that there is uncertaintity in the classes at each time point, it assumes that the predicted classes are known exactly. The consequence of this will be to produce final results where the standard errors are smaller than they should be. LMest is based on the likelihood for the complete model, and so does not have this problem. While the results here are similar to the previous publication, it is not assured. LMest would also allow the number of classes to be determined using all data.

If the paper is viewed as a tutorial, it needs to be more explanatory. The data format and the commands should be described. If it isn't a tutorial it doesn't seem to be of much interest, as it only repeats a previously published analysis.

Bartolucci, F., Pandolfi, S., & Pennoni, F. (2017). LMest: An R Package for Latent Markov Models for Longitudinal Categorical Data. Journal of Statistical Software, 81(4), 1–38. https://doi.org/10.18637/jss.v081.i04

Reviewer #3: The authors present a re-analysis of a dataset that was previously analyzed using LTA. In this paper. The authors show how the original results, which were obtained using Mplus (a commercial software package) can be replicated using R. The authors start with a short introduction of the LTA technique, followed by a short description of the used data and their followed approach. This is followed by a description of the results and a discussion that reflects on the strengths and limitations of the proposed alternative approach.

Generally, the article is clearly structured and written and the proposed approach seems sound. I like initiatives that focus on making statistical techniques available in R and applaud the authors' efforts. However, I did have some difficulty understanding what the author’s primary objective is with the paper. On the one hand, the objective seems to demonstrate that results that were obtained using one type of software can be replicated using similar analytical steps using other software in the same dataset. On the other hand, the objective seems to provide a framework for conducting LTA in R and to demonstrate how LTA can be carried out (i.e. a tutorial paper). I hope the paper is intended as a tutorial, as I personally see much more interest and added value in that. However, in that case, I would expect a more detailed description of (A) how each step can be carried out (including code examples) and (B) what considerations play a role in the multiple decisions that are made during the analysis process.

Below I provide some more specific comments and questions.

Introduction

• The authors state that: ‘Consequently, LTA is suitable for answering other research questions than the fixed membership approaches outlined above’, but do not really go into what defines the kinds of research questions that can be answered with LTA and what differentiates those RQs from those that can be answered using growth-based mixture models (e.g., GMM, LCGA, HMM). For researchers that are still trying to determine if LTA is suitable for their needs, it might be helpful to add this information.

• The added value of the proposed approach vs. ‘lmest’ is not yet made very clear in the introduction.

Methods

• It would be useful if the used analytical procedures and results of the original study were shortly summarized, so readers can actually see to what extent the current results are in line with the original ones.

• Estimation of an LCA at each time-point separately in step 1 (prior to the transition analysis) seems like an inuitive choice, but it does rest on a strong assumption of measurement invariance that can only be checked after estimation. This does raise two questions (1) What if measurement invariance (i.e. thresholds) does not hold (e.g., what if statistical fit-indices show different class-numbers to be optimal or different model parameter estimates?). (2) Why do the authors not strive to estimate a measurement invariant model to begin with (basically, this would equate to applying LCA to both time-points with equality constraints on the item thresholds). Their approach may be more flexible, but may also lead to theoretical and practical issues. In either case. In order for their approach to be useful to others and to make it a viable alternative for commercial packages, the authors need to pay more attention to this aspect of LTA in their manuscript. It is shortly mentioned in the discussion, but practical and theoretical implications for their LTA approach remain unclear.

• The description of the LCA fitting-process and models selection could be more detailed, as to provide more guidance to prospective LTA users. For instance: how do you use AIC and BIC to select the optimal model? How to select the optimal model given the data in border cases where models show ICs that hardly differ? How to proceed when BIC and AIC are inconclusive (e.g., keep decreasing with each class addition)? What to do if BIC and AIC contradict each other; how should we weight them in our final decision? If practical/pragmatic criteria are used, what should those be (i.e. usable class sizes vs. interesting differentiation of class characteristics) and how should those be prioritized? As it stands, the described model-selection procedure as well as the presented results could strike the reader as rather arbitrary.

• How should users handle the risk of solutions at a local maximum?

• It is not clear why logistic regression is chosen in favor of multinomial regression in step 2. Why is this necessary? The authors state that this entails only a minor sacrifice of efficiency, but it is unclear what the actual implications are for the analyses, especially given the fact that logistic regression can easily become (severely) underpowered with small cell-counts and inclusion of interaction terms. The authors also refer to the advantage of not having to make proportionality assumptions with their approach. However, as far as I am aware such assumptions (e.g., proportional odds) are generally not made in multinomial regression methods for nominal variables. Could the authors eleborate more on their reasons?

• There is very little information about the used imputation method. For future users of the approach, it could be helpful to add some infocmation about how this was set up. What imputation algorithm was used for the different types of variables, including latent-class variable? How many iterations are used? How is the predictor matrix configured? Etc. There is also no information about the pooling procedures (e.g., application of Rubin’s rules). It might be useful to add this as well

• Version numbers of the used R-packages are not mentioned in the manuscript.

Results

• The selection of the optimal model did strike me as rather ad hoc (see above). For instance, the criterion of ‘around or less than 10’ is invoked to base the final selection on, but this was not mentioned previously. I also found it confusing that the 3-class model was referred to as ‘a parsimonious choice’, given that not the 3-class, but the 2-class model had the lowest values on the more conservative BIC at both time-points. It would be helpful if the reader understood how the AIC and BIC values were weighted as two sources of information in the final decision to select the 3-class model.

• The Step 3 analyses were probably underpowered. It might be a good idea to explore the actually available statistical power and reflect on this in the discussion section. This could also help prospective users to determine if their sample size is sufficient to begin with.

Discussion

• The second paragraph lists an interesting array of possibilities for conducting LTA (or spin-off techniques) within the provided framework (e.g., ML approach, suitable imputation algorithms). Here, some of the added value of the proposed R-approach becomes clear. Here, it also mentioned that classes do not need to be constant across time, but that does beg the question of how much value the authors place on measurement invariance and its key-role in the interpretation of LTA models. Do they propose a different type of model/interpretation?

• It is not clear how the use of a single multinomial regression (e.g., with package ‘nnet’) leads to more programming complexity than the use of multiple logistic regressions.

6. PLOS authors have the option to publish the peer review history of their article (what does this mean?). If published, this will include your full peer review and any attached files.

Reviewer #1: **Yes: **Daniel Gerhard

Reviewer #2: No

Reviewer #3: **Yes: **Klaas J. Wardenaar

---

## [Author Response · Author response to Decision Letter 0]

20 Jun 2024

Rebuttal letter

Reviewer #1

The authors describe an implementation of latent transition analysis (LTA) with the statistical software R, based on an example study about wellbeing of high school students previously published in 2022.

Instead of parameter estimation in a single step, the analysis is split into at least two steps, first estimating latent classes, and then estimating transition probabilities and the effect of predictor variables on these transitions in a second or third step.

Given that the latent class membership has been estimated in Step 1, there is uncertainty about whether a participant belongs to a particular class. The logistic regression models assume instead a fixed class membership for each participant. Is it possible that this assumption affects the results? Can the uncertainty about class membership be carried forward into Step 2, for example using weights based on the item-response probabilities?

Reply: Thanks for providing such an interesting idea. To our knowledge none of the statistical software packages used for LTA takes this uncertainty into account. Moreover, it seems from the literature (on applying LCA for defining clusters to be used in subsequent statistical analyses) that this uncertainty is often entirely ignored. However, we feel that exploring this idea in detail is beyond the scope of the present tutorial (although some pointers to a number of papers are provided). This limitation now features prominently in the discussion. 

Step 2 and Step 3 have been simplified by using separate binary logistic regression models instead of a multinomial regression model. If the correlation between classes is not important, this is a reasonable approximation, but the authors note in the discussion that multinomial regression models are a feasible alternative. I agree that this simplification also allows for modelling more complex sampling structures, although it might be more of a challenge to include them in the first LCA step.

Reply: Thanks for this great comment. We've clarified the point about correlation in the discussion. Also, in the discussion we now mention that more complex study designs may lead to more challenges for the LCA in Step 1 and provide some ideas for solutions.

In the R script, it looks as if separate models were fitted for each predictor variable. Can we instead assume a single model conditioning on the full set of predictors?

Reply: Thanks for this great comment and for going through the R script. We've now spelled out in the Methods section that separate models were fitted for each predictor. In principle, it would be possible to include multiple interaction terms in the same logistic regression model, but in practice it will most likely rapidly lead to convergence problems. Also, current practice in epidemiology when exploring effect modification is to look at one effect modificator at a time as it allows for a straightforward interpretation.

As this is a replicated analysis of the Kassis et al. 2022 study, would it be possible to show the differences to the previous analysis in the Results Section?

Reply: Thanks for the comment. We've now summarized findings from the original paper alongside our findings throughout the Results section whenever possible, implying that we now report both AIC and the adjusted BIC for our approach and some tables show results from our approach and from the original study for easy comparison. Likewise, more details on the methods used in the original study are provided in the Methods section.

Minor issues:

Page 8: There is some functionality for carrying out LTA is available in R [12].

Instead: There is some functionality available in R [12] for carrying out LTA.

Page 8: Missing empty space: Likert scales were originally used, butonly dichotomized versions

Checking capital letters for names, for example, Wave 1 and 2

Reply: Thanks for spotting these mistakes. Now fixed.

Reviewer #2

The title is slightly misleading. Subgroup is usually used to refer to groups of subjects, that is the group of subjects that the subject is in for all time, which in this case there is only one. Then the subject may change class at each time point.

Reply: You're certainly right. Thanks for pointing out. We've reworded the title and the text, replacing "subgroup" by latent class, which actually led to a much more coherent and consistent manuscript text as latent classes were already used in some places.

The introduction implied that the paper was going to provide a tutorial on using the LMest package to fit LTA models. Although it may not be suitable for PLOS, this would be useful, as it is a model that is not covered in the authors description of the LMEst package. (Bartolucci et al, 2017). Note that it is a simplified model to that shown in their 6.2, with only one class for individuals, so may be fitted with lmest.

Reply: Thanks for the comment. We agree that we weren't sufficiently precise in the abstract and introduction. We've now clarified that our approach relies on combining latent class analysis and logistic regression (or, more generally, generalized linear models for binary data with arbitrary link functions and various extensions/modifications of these models), as a flexible alternative to multinomial models using odds ratio as effect measure. 

However they show how to fit these models in a 3 step process. I think most statisticians would find this unappealing. When determining the latent classes, this requires them to be determined at each wave separately, rather than having a common set of classes, which are applied at all waves. The major difficulty is that rather than considering that there is uncertaintity in the classes at each time point, it assumes that the predicted classes are known exactly. The consequence of this will be to produce final results where the standard errors are smaller than they should be. LMest is based on the likelihood for the complete model, and so does not have this problem. While the results here are similar to the previous publication, it is not assured. LMest would also allow the number of classes to be determined using all data.

Reply: Thanks for this very good comment. We certainly agree that some uncertainty might not be propagated from Step 1 to the subsequent Step 2 and Step 3. This issue was also pointed out by Reviewer #1. This limitation now features prominently in the discussion. We also suggest some ideas for addressing this shortcoming within the proposed modelling framework.

If the paper is viewed as a tutorial, it needs to be more explanatory. The data format and the commands should be described. If it isn't a tutorial it doesn't seem to be of much interest, as it only repeats a previously published analysis.

Reply: Thanks for this useful comment. We've now included small code snipppets with the functions used in the manuscript to show how to do it in practice, in much the same way as for other tutorials in statistical journals such as Statistics in Medicine. We've also changed the title to better reflect that it's a tutorial.

Please also note that the results on how transitions depend on various baseline characteristics are novel as they were not part of the original publication by Kassis et al. where no such statistical analyses were carried out (they only carried out cross-sectional analyses wave by wave).

Reviewer #3

The authors present a re-analysis of a dataset that was previously analyzed using LTA. In this paper. The authors show how the original results, which were obtained using Mplus (a commercial software package) can be replicated using R. The authors start with a short introduction of the LTA technique, followed by a short description of the used data and their followed approach. This is followed by a description of the results and a discussion that reflects on the strengths and limitations of the proposed alternative approach.

Generally, the article is clearly structured and written and the proposed approach seems sound. I like initiatives that focus on making statistical techniques available in R and applaud the authors' efforts. However, I did have some difficulty understanding what the author’s primary objective is with the paper. On the one hand, the objective seems to demonstrate that results that were obtained using one type of software can be replicated using similar analytical steps using other software in the same dataset. On the other hand, the objective seems to provide a framework for conducting LTA in R and to demonstrate how LTA can be carried out (i.e. a tutorial paper). I hope the paper is intended as a tutorial, as I personally see much more interest and added value in that. However, in that case, I would expect a more detailed description of (A) how each step can be carried out (including code examples) and (B) what considerations play a role in the multiple decisions that are made during the analysis process.

Reply: Thanks for this very useful comment. The paper is intended to be a tutorial. We've attempted to make this more clear from the introduction. It's now also reflected in the revised title.

Additionally, in the Results section, we've now included small code snipppets in the manuscript, illustrating the use of R functions to show how to do it in practice, in much the same way as for other tutorials as seen in statistical journals (such as Statistics in Medicine). We've also attempted to provide some guidance on the decisions involved in the modelling steps.

Below I provide some more specific comments and questions.

Reply: Thanks for providing these excellent and detailed comments and suggestions. Much appreciated as overall they led to a much more clean and structured manuscript with methodology introduced in the Methods section and results in the Results section.

Introduction

• The authors state that: ‘Consequently, LTA is suitable for answering other research questions than the fixed membership approaches outlined above’, but do not really go into what defines the kinds of research questions that can be answered with LTA and what differentiates those RQs from those that can be answered using growth-based mixture models (e.g., GMM, LCGA, HMM). For researchers that are still trying to determine if LTA is suitable for their needs, it might be helpful to add this information.

Reply: We've provided more details, exemplifying where LTA makes sense to use.

• The added value of the proposed approach vs. ‘lmest’ is not yet made very clear in the introduction.

Reply: You're right. Thanks for noticing. We've now pointed out that "LMest", like commercial software, relies on multinomial models with odds ratios as the effect measure; please also see our reply below on multinomial models.

Methods

• It would be useful if the used analytical procedures and results of the original study were shortly summarized, so readers can actually see to what extent the current results are in line with the original ones.

Reply: Thanks for making this point (which was also made by Reviewer 1). We've summarized findings from the original paper alongside our findings throughout the Results section whenever possible; some tables now also features findings from the original study to make comparisons easier. Likewise, more details on the methods used in the original study are provided in the Methods section.

• Estimation of an LCA at each time-point separately in step 1 (prior to the transition analysis) seems like an inuitive choice, but it does rest on a strong assumption of measurement invariance that can only be checked after estimation. This does raise two questions (1) What if measurement invariance (i.e. thresholds) does not hold (e.g., what if statistical fit-indices show different class-numbers to be optimal or different model parameter estimates?). (2) Why do the authors not strive to estimate a measurement invariant model to begin with (basically, this would equate to applying LCA to both time-points with equality constraints on the item thresholds). Their approach may be more flexible, but may also lead to theoretical and practical issues. In either case. In order for their approach to be useful to others and to make it a viable alternative for commercial packages, the authors need to pay more attention to this aspect of LTA in their manuscript. It is shortly mentioned in the discussion, but practical and theoretical implications for their LTA approach remain unclear.

Reply: Thanks for this interesting point. Why is measurement invariance needed in the first place? If latent classes changes somewhat over time then it may still be interesting to describe transitions between classes; the notion of a transition does not require measurement invariance. Of course it makes life easier in terms of interpretation, but it's not always that life is so simple. The approach that we propose has no such restriction and we believe this is a strength. If it's possible after model fitting, by some means, to show that measurement invariance is likely the case it's fine but if not it's also not a problem. 

Also, absence of measurement variance is a problem in case you want to fit multinomial models, but not in case the analyses are split into a number of logistic regression models. We've attempted to showcast this very important point more in the discussion. Thanks again for the input.

Moreover, there is nowadays a general tendency in statistical practice to narrow down the use of p-values, abandoning the use of statistical test procedures for assessing model assumptions as it is essentially the sample size that determines whether an assumption can be accepted or not, implicitly leading to a preference for smaller studies with a lot of variation as then the p-value more easily becomes non-significant (interest lies in not rejecting the null hypothesis, which is not how statistical hypothesis testing was intended to be used in the first place). 

• The description of the LCA fitting-process and models selection could be more detailed, as to provide more guidance to prospective LTA users. For instance: how do you use AIC and BIC to select the optimal model? How to select the optimal model given the data in border cases where models show ICs that hardly differ? How to proceed when BIC and AIC are inconclusive (e.g., keep decreasing with each class addition)? What to do if BIC and AIC contradict each other; how should we weight them in our final decision? If practical/pragmatic criteria are used, what should those be (i.e. usable class sizes vs. interesting differentiation of class characteristics) and how should those be prioritized? As it stands, the described model-selection procedure as well as the presented results could strike the reader as rather arbitrary.

• How should users handle the risk of solutions at a local maximum?

Reply: Thanks for the comments on LCA. We've included more details on LCA, but to avoid that the manuscript drifts too much away from LTA and towards how to use LCA in general, we've also provided references as pointers in some places.

• It is not clear why logistic regression is chosen in favor of multinomial regression in step 2. Why is this necessary? The authors state that this entails only a minor sacrifice of efficiency, but it is unclear what the actual implications are for the analyses, especially given the fact that logistic regression can easily become (severely) underpowered with small cell-counts and inclusion of interaction terms. The authors also refer to the advantage of not having to make proportionality assumptions with their approach. However, as far as I am aware such assumptions (e.g., proportional odds) are generally not made in multinomial regression methods for nominal variables. Could the authors eleborate more on their reasons?

Reply: It's true that multinomial models may have less problems than logistic regression in case of sparse data, but it comes at the price of stronger assumptions. It should also be noted that, in practice, studies where LTA is used often aren't so small and small latent classes aren't that frequent after having applied a criterion such as AIC or BIC. However, this is indeed a limitation of the proposed approach and we've included it in the discussion.

• There is very little information a

---

## [Decision Letter · Decision Letter 1]

16 Jul 2024

PONE-D-24-09372R1Flexible and modular latent transition analysis - a tutorial using RPLOS ONE

Dear Dr. Ritz,

Thank you for submitting your manuscript to PLOS ONE. After careful consideration, we feel that it has merit but does not fully meet PLOS ONE’s publication criteria as it currently stands. Therefore, we invite you to submit a revised version of the manuscript that addresses the points raised during the review process. Please submit your revised manuscript by Aug 30 2024 11:59PM. If you will need more time than this to complete your revisions, please reply to this message or contact the journal office at plosone@plos.org. Please include the following items when submitting your revised manuscript:A rebuttal letter that responds to each point raised by the academic editor and reviewer(s). You should upload this letter as a separate file labeled 'Response to Reviewers'.A marked-up copy of your manuscript that highlights changes made to the original version. You should upload this as a separate file labeled 'Revised Manuscript with Track Changes'.An unmarked version of your revised paper without tracked changes. You should upload this as a separate file labeled 'Manuscript'.If applicable, we recommend that you deposit your laboratory protocols in protocols.io to enhance the reproducibility of your results. Protocols.io assigns your protocol its own identifier (DOI) so that it can be cited independently in the future. For instructions see: https://journals.plos.org/plosone/s/submission-guidelines#loc-laboratory-protocols. Additionally, PLOS ONE offers an option for publishing peer-reviewed Lab Protocol articles, which describe protocols hosted on protocols.io. Read more information on sharing protocols at https://plos.org/protocols?utm_medium=editorial-email&utm_source=authorletters&utm_campaign=protocols.

We look forward to receiving your revised manuscript.

Kind regards,

Sandar Tin Tin

Academic Editor

PLOS ONE

Journal Requirements:

Reviewers' comments:

Reviewer's Responses to Questions

**Comments to the Author**

1. If the authors have adequately addressed your comments raised in a previous round of review and you feel that this manuscript is now acceptable for publication, you may indicate that here to bypass the “Comments to the Author” section, enter your conflict of interest statement in the “Confidential to Editor” section, and submit your "Accept" recommendation.

Reviewer #1: All comments have been addressed

Reviewer #3: (No Response)

2. Is the manuscript technically sound, and do the data support the conclusions?

Reviewer #1: Yes

Reviewer #3: Yes

3. Has the statistical analysis been performed appropriately and rigorously? 

Reviewer #1: Yes

Reviewer #3: Yes

4. Have the authors made all data underlying the findings in their manuscript fully available?

Reviewer #1: Yes

Reviewer #3: Yes

5. Is the manuscript presented in an intelligible fashion and written in standard English?

Reviewer #1: Yes

Reviewer #3: Yes

6. Review Comments to the Author

Reviewer #1: (No Response)

Reviewer #3: I read the revised manuscript and was happy to see that it is now much clearer that the objective of the paper is to provide a tutorial for conducting LTA in R. Generally, I found that most of my previous comments were addressed well by the authors. The resulting manuscript is nearing completion, but I did have some remaining points. Nothing major, though.

• I agree with the authors that measurement invariance is and should not be a very strict requirement when doing LTA (although LTA users should be aware of it, which they not always are…) and that overreliance on significance tests to ‘establish’ measurement non-invariance is often not a good idea. However, I think that some references are needed to back this latter point up for MI-tests specifically, instead of presenting general statements about assumption tests being unsuitable in any application. This will provide future LTA users with bit more formal guidance.

• I would not consider the use of random starts as optional (and ten random starts strikes me as rather small number), especially when estimating more complex models. I understand that the focus of the tutorial is on the LTA part and that the importance of random starts may be assumed to be known as general LCA knowledge. However, working with model solutions at local maxima is a serious risk in LTA, so I would stress the importance of always thoroughly checking this and not only ‘when suspected’ (after all: based on what concrete clues should such suspicions arise?).

• I previously expressed doubt about the statement that using multiple logistic regression models addresses the issue of making a proportionality assumption in traditional LTA, as I do not see how multinomial regression makes strict assumptions about this. Proportional odds are assumed in ordinal regression, but that is not what is used as a standard in LTA, as far as I’m aware. This point was not really addressed in the response or the manuscript. Maybe I misunderstand what you mean here. Could you elaborate on and/or clarify this point?

• Although I still have some reservations about the added value of using multiple logistic regression models instead of a single multinomial regression model to calculate transition probabilities, I do feel that the authors do address this sufficiently throughout the manuscript for readers to make up their mind about this. My only remaining question then is how the authors position their approach vs. the ‘LMest’ package. They do not explicitly mention this (e.g., in the introduction), whereas this would be very informative for readers that need to decide what approach to use when they need to stick to R (I for instance like how this is usually done in J Stat Software tutorial papers).

• By positioning their approach as a ‘3-step’ approach, the authors do raise the question how their approach relates to well-known previously described general 3-step approaches to LCA (e.g., Vermunt [2010; https://doi.org/10.1093/pan/mpq025]). For completeness, the authors may want to reflect on this.

Minor remarks

• I noticed that some of the additions and changes in the text led to hard to read or grammatically suboptimal sentences (e.g., first sentence of abstract: ‘Latent transition analysis (LTA) is a useful statistical modelling approach for identifying latent classes and describe subsequent transitions between these classes over time’). I spotted similar small issues in some other parts of the text.

• In the first paragraph of the introduction, a distinction is made between ‘ones that focus on characterization of determinants for belonging to certain longitudinal patterns and ones that focus on characterization of determinants of transitions between longitudinal patterns’. The latter part of this sentence is a bit confusing, given that in LTA the patterns themselves are not longitudinal, but merely the transitions between these patterns. You could consider rephrasing this slightly.

• You could consider adding an illustration of how to set the seed for the random number generator in Box 2.

• In Box 3, the model contains the term ‘lca1:Age-Age’ Is this correct? I did not understand why ‘Age-Age’ rather than ‘Age’ was included here and it was not explained in the text.

• It might be useful to mention that the R-code to test the other covariates’ effects shown in Table 3 is given in the appendix.

7. PLOS authors have the option to publish the peer review history of their article (what does this mean?). If published, this will include your full peer review and any attached files.

Reviewer #1: **Yes: **Daniel Gerhard

Reviewer #3: **Yes: **Klaas J. Wardenaar

---

## [Author Response · Author response to Decision Letter 1]

16 Aug 2024

Rebuttal letter - revision 2

Reviewer #2

I read the revised manuscript and was happy to see that it is now much clearer that the objective of the paper is to provide a tutorial for conducting LTA in R. Generally, I found that most of my previous comments were addressed well by the authors. The resulting manuscript is nearing completion, but I did have some remaining points. Nothing major, though.

Reply: Thank you so much once more for the careful reading and for the useful comments and inputs. Please see our replies below.

• I agree with the authors that measurement invariance is and should not be a very strict requirement when doing LTA (although LTA users should be aware of it, which they not always are…) and that overreliance on significance tests to ‘establish’ measurement non-invariance is often not a good idea. However, I think that some references are needed to back this latter point up for MI-tests specifically, instead of presenting general statements about assumption tests being unsuitable in any application. This will provide future LTA users with bit more formal guidance.

Reply: Thanks for mentioning. We realized that this is a quite extensively debated topic! We now refer to the papers by Schmitt & Kuljanin and Welzel et al. who acknowledge the subjective element in the assessment of measurement invariance and downplay the role of statistical tests. 

• I would not consider the use of random starts as optional (and ten random starts strikes me as rather small number), especially when estimating more complex models. I understand that the focus of the tutorial is on the LTA part and that the importance of random starts may be assumed to be known as general LCA knowledge. However, working with model solutions at local maxima is a serious risk in LTA, so I would stress the importance of always thoroughly checking this and not only ‘when suspected’ (after all: based on what concrete clues should such suspicions arise?).

Reply: Thanks for pointing out. A good point. We've rephrased the text to stress that using multiple starting values should be done routinely. Also, we re-ran the R code now including the "nrep = 10" argument so that 10 random starting values were used. For wave 1 and 5 latent classes the information criteria changed a little but all other results remained unaltered (Table 1 was updated accordingly). The supplementary material on zenodo.org was also updated accordingly.

• I previously expressed doubt about the statement that using multiple logistic regression models addresses the issue of making a proportionality assumption in traditional LTA, as I do not see how multinomial regression makes strict assumptions about this. Proportional odds are assumed in ordinal regression, but that is not what is used as a standard in LTA, as far as I’m aware. This point was not really addressed in the response or the manuscript. Maybe I misunderstand what you mean here. Could you elaborate on and/or clarify this point?

Reply: You're right that it's not a general assumption. We'd a specific multinomial model in mind but unfortunately not the general case. The text has been revised omitting such limiting but irrelevant model assumptions.

• Although I still have some reservations about the added value of using multiple logistic regression models instead of a single multinomial regression model to calculate transition probabilities, I do feel that the authors do address this sufficiently throughout the manuscript for readers to make up their mind about this. My only remaining question then is how the authors position their approach vs. the ‘LMest’ package. They do not explicitly mention this (e.g., in the introduction), whereas this would be very informative for readers that need to decide what approach to use when they need to stick to R (I for instance like how this is usually done in J Stat Software tutorial papers).

Reply: This is a good point that we perhaps had not addressed in sufficient detail previously. From our point of view the package LMest is of limited value for analyzing more complex study designs involving hierarchical structures; indeed this was part of the initial motivation for the present work. Also, it lacks flexible handling of missing values and is limited to odds ratios as effect measures. So in the end it depends on the complexity of the LTA to be undertaken and the desired output format. We've updated the discussion accordingly.

• By positioning their approach as a ‘3-step’ approach, the authors do raise the question how their approach relates to well-known previously described general 3-step approaches to LCA (e.g., Vermunt [2010; https://doi.org/10.1093/pan/mpq025]). For completeness, the authors may want to reflect on this.

Reply: Thanks for this very interesting paper. In principle the proposed weighted extension proposed in the paper could also be used for our approach as a means to properly propagate uncertainty from one step to the next. We now mention this great idea, which is yet another modular extension feasible using R packages, in the discussion.

Minor remarks

• I noticed that some of the additions and changes in the text led to hard to read or grammatically suboptimal sentences (e.g., first sentence of abstract: ‘Latent transition analysis (LTA) is a useful statistical modelling approach for identifying latent classes and describe subsequent transitions between these classes over time’). I spotted similar small issues in some other parts of the text.

Reply: We've been going through the manuscript and improved a number of longer and hard-to-read sentences.

• In the first paragraph of the introduction, a distinction is made between ‘ones that focus on characterization of determinants for belonging to certain longitudinal patterns and ones that focus on characterization of determinants of transitions between longitudinal patterns’. The latter part of this sentence is a bit confusing, given that in LTA the patterns themselves are not longitudinal, but merely the transitions between these patterns. You could consider rephrasing this slightly.

Reply: Point taken. We've revised the first paragraph somewhat to make the difference more clear.

• You could consider adding an illustration of how to set the seed for the random number generator in Box 2.

Reply: Thanks. This is an important detail. We've included the R command in both Box 1 and 2.

• In Box 3, the model contains the term ‘lca1:Age-Age’ Is this correct? I did not understand why ‘Age-Age’ rather than ‘Age’ was included here and it was not explained in the text.

Reply: Thank you for noting. The "-Age" is included to remove the use of a reference group, which is otherwise the default behaviour in R, such that estimates per group (in this latent class) are obtained directly. We've updated the supplementary material where the more detailed R-specific details are explained.

• It might be useful to mention that the R-code to test the other covariates’ effects shown in Table 3 is given in the appendix.

Reply: Thanks. Such a sentence is now included in the manuscript.

---

## [Editor Report · Decision Letter 2]

2 Jan 2025

Flexible and modular latent transition analysis - a tutorial using R

PONE-D-24-09372R2

Dear Dr. Author ,

We’re pleased to inform you that your manuscript has been judged scientifically suitable for publication and will be formally accepted for publication once it meets all outstanding technical requirements.

Kind regards,

Umair Khalil, PhD

Academic Editor

PLOS ONE

Additional Editor Comments (optional):

please include the R codes which you used to get the results
---

## [Editor Report · Acceptance letter]

8 Jan 2025

PONE-D-24-09372R2 

PLOS ONE

Dear Dr. Ritz, 

I'm pleased to inform you that your manuscript has been deemed suitable for publication in PLOS ONE. Congratulations! Your manuscript is now being handed over to our production team.

Kind regards, 

on behalf of

Dr. Umair Khalil 

Academic Editor

PLOS ONE